# LEARNING PARAMETRISED GRAPH SHIFT OPERATORS

**George Dasoulas**[*12], **Johannes F. Lutzeyer**[*1] **& Michalis Vazirgiannis**[1]

[1] DaSciM, LIX, École Polytechnique, Institute Polytechnique de Paris, France

[2] Noah's Ark Lab, Huawei Technologies France

{georgios.dasoulas, johannes.lutzeyer}@polytechnique.edu,
mvazirg@lix.polytechnique.fr

## ABSTRACT

In many domains data is currently represented as graphs and therefore, the graph representation of this data becomes increasingly important in machine learning. Network data is, implicitly or explicitly, always represented using a graph shift operator (GSO) with the most common choices being the adjacency, Laplacian matrices and their normalisations. In this paper, a novel parametrised GSO (PGSO) is proposed, where specific parameter values result in the most commonly used GSOs and message-passing operators in graph neural network (GNN) frameworks. The PGSO is suggested as a replacement of the standard GSOs that are used in state-of-the-art GNN architectures and the optimisation of the PGSO parameters is seamlessly included in the model training. It is proved that the PGSO has real eigenvalues and a set of real eigenvectors independent of the parameter values and spectral bounds on the PGSO are derived. PGSO parameters are shown to adapt to the sparsity of the graph structure in a study on stochastic blockmodel networks, where they are found to automatically replicate the GSO regularisation found in the literature. On several real-world datasets the accuracy of state-of-the-art GNN architectures is improved by the inclusion of the PGSO in both node- and graph-classification tasks.

## 1 INTRODUCTION

Real-world data and applications often involve significant structural complexity and as a consequence graph representation learning attracts great research interest (Hamilton et al., 2017b; Wu et al., 2020). The topology of the observations plays a central role when performing machine learning tasks on graph structured data. A variety of supervised, semi-supervised or unsupervised graph learning algorithms employ different forms of operators that encode the topology of these observations. The most commonly used operators are the adjacency matrix, the Laplacian matrix and their normalised variants. All of these matrices belong to a general set of linear operators, the *Graph Shift Operators (GSOs)* (Sandryhaila & Moura, 2013; Mateos et al., 2019).

Graph Neural Networks (GNNs), the main application domain in this paper, are representative cases of algorithms that use chosen GSOs to encode the graph structure, i.e., to encode neighbourhoods used in the aggregation operators. Several GNN models (Kipf & Welling, 2017; Hamilton et al., 2017a; Xu et al., 2019) choose different variants of normalised adjacency matrices as GSOs. Interestingly, in a variety of tasks and datasets, the incorporation of explicit structural information of neighbourhoods into the model is found to improve results (Pei et al., 2020; Zhang & Chen, 2018; You et al., 2019), leading us to conclude that the chosen GSO is not entirely capturing the information of the data topology. In most of these approaches, the GSO is chosen without an analysis of the impact of this choice of representation. From this observation arise our two research questions.

**Question 1:** *Is there a single optimal representation to encode graph structures or is the optimal representation task- and data-dependent?*

On different tasks and datasets, the choice between the different representations encoded by the different graph shift operator matrices has shown to be a consequential decision. Due to the past

---

*Equal contribution.

successful approaches that use different GSOs for different tasks and datasets, it is natural to assume that there is no single optimal representation for all scenarios. Finding an optimal representation of network data could contribute positively to a range of learning tasks such as node and graph classification or community detection. Fundamental to this search is an answer to Question 1. In addition, we pose the following second research question.

**Question 2:** *Can we learn such an optimal representation to encode graph structure in a numerically stable and computationally efficient way?*

The utilisation of a GSO as a topology representation is currently a hand-engineered choice of normalised variants of the adjacency matrix. Thus, the learnable representation of node interactions is transferred into either convolutional filters (Kipf & Welling, 2017; Hamilton et al., 2017a) or attention weights (Veličković et al., 2018), keeping the used GSO constant. In this work, we suggest a parametrisation of the GSO. Specific parameter values in our proposed parametrised (and differentiable) GSO result in the most commonly used GSOs, namely the adjacency, unnormalised Laplacian and both normalised Laplacian matrices, and GNN aggregation functions, e.g., the averaging and summation message passing operations. The beauty of this innovation is that it can be seamlessly included in both message passing and convolutional GNN architectures. Optimising the operator parameters will allow us to find answers to our two research questions.

The remainder of this paper is organised as follows. In Section 2, we give an overview of related work in the literature. Then in Section 3, we define our parametrised graph shift operator (PGSO) and discuss how it can be incorporated into many state-of-the-art GNN architectures. This is followed by a spectral analysis of our PGSO in Section 4, where we observe good numerical stability in practice. In Section 5, we analyse the performance of GNN architectures augmented by the PGSO in a node classification task on a set of stochastic blockmodel graphs with varying sparsity and on learning tasks performed on several real-world datasets.

## 2  RELATED WORK

GSOs emerge in different research fields such as in physics, network science, computer science and mathematics, taking usually the form of either graph Laplacian normalisations or variants of the adjacency matrix. In an abundant number of machine learning applications the expressivity of GSOs is exploited, e.g., in unsupervised learning (von Luxburg, 2007; Kim et al., 2008), semi-supervised node classification on graph-structured data (Kipf & Welling, 2017; Schlichtkrull et al., 2018) and supervised learning on computer vision tasks (Chang & Yeung, 2006). The majority of these works assumes a specified normalised version of the Laplacian that encodes the structural information of the problem and usually these versions differ depending on the analysed dataset and the end-user task. Recently, new findings on the impact of the chosen Laplacian representation have emerged that highlight the contribution of Laplacian regularisation (Dall'Amico et al., 2020; Saade et al., 2014; Dall'Amico et al., 2019). The different GSO choices in different tasks indicate a data-dependent relation between the structure of the data and its optimal GSO representation. This observation motivates us to investigate how beneficial a well-chosen GSO can be for a learning task on structured data.

GNNs use a variety of GSOs to encode neighbourhood topologies, either normalisations of the adjacency matrix (Xu et al., 2019; Hamilton et al., 2017a) or normalisations of the graph Laplacian (Kipf & Welling, 2017; Wu et al., 2019). Due to the efficiency and the predictive performance of GNNs, a research interest has recently emerged in their expressive power. One of the examined aspects is that of the equivalence of the GNNs' expressive power with that of the Weisfeiler-Lehman graph isomorphism test (Dasoulas et al., 2020; Maron et al., 2019; Morris et al., 2019; Xu et al., 2019). Another research direction is that of analysing the depth and the width of GNNs, moving one step forward to the design of deep GNNs (Loukas, 2020; Li et al., 2018; Liu et al., 2020; Alon & Yahav, 2020). In this analysis, the authors study phenomena of Laplacian oversmoothing and combinatorial oversquashing, that harm the expressiveness of GNNs. In most of these approaches, however, the used GSO is fixed without a motivation of the choice. We hope that the parametrised GSO that is presented in this work can contribute positively to the expressivity analysis of GNNs.

We will now delineate our approach and that of a closely related work by Klicpera et al. (2019). Klicpera et al. (2019) demonstrate that varying the choice of the GSO in the message passing step

of GNNs can lead to significant performance gains. In Klicpera et al. (2019) two fixed diffusion operators with a much larger receptive field than the 1-hop neighbourhood convolutions, are inserted into the architectures, leading to a significant improvement of the GNNs' performance. In our work here we replace the GSOs in GNN frameworks with the PGSO, which has a receptive field equal to the 1-hop neighbourhood of the nodes. We find that parameter values of our PGSO can be trained in a numerically stable fashion, which allows us to chose a parametric form unifying the most common GSOs and aggregation functions. As with standard GNN architectures the receptive field of the convolutions is increased in our architectures by stacking additional layers. Klicpera et al. (2019) increase the size of the receptive field and keep the neighbourhood representation fixed, while we keep the size of the receptive field fixed and learn the neighbourhood representation.

## 3 Parametrised Graph Shift Operators

We define notation and fundamental concepts in Section 3.1 and introduce our proposed parametrised graph shift operator $\gamma(A, \mathcal{S})$ in Section 3.2. In Section 3.3, we provide a detailed discussion of how $\gamma(A, \mathcal{S})$ can be applied to the message-passing operation in GNNs and we demonstrate use cases of the incorporation of $\gamma(A, \mathcal{S})$ in different GNN architectures.

### 3.1 Preliminaries

Let a graph $G$ be a tuple, $G = (V, E)$, where $V$ and $E$ are the sets of nodes and edges and let $|V| = n$. We assume the graph $G$ to be *attributed* with attribute matrix $X \in \mathbb{R}^{n \times d}$, where the $i^{\text{th}}$ row of $X$ contains the $d$-dimensional attribute vector corresponding to node $v_i$. We denote the $n \times n$ identity matrix by $I_n$ and the $n$-dimensional column vector of all ones by $\mathbf{1}_n$. Given the node and edge sets, one can define the *adjacency matrix*, denoted $A \in [0, 1]^{n \times n}$, where $A_{ij} \neq 0$ if and only if $(i, j) \in E$, and the degree matrix of $A$ as $D = \text{Diag}(A\mathbf{1}_n)$.

Recently the notion of a GSO has been defined as a general family of operators which enable the propagation of signals over graph structures (Sandryhaila & Moura, 2013; Shuman et al., 2013).

**Definition 1. Graph Shift Operator** A matrix $S \in \mathbb{R}^{n \times n}$ is called a *Graph Shift Operator* (GSO) if it satisfies $S_{ij} = 0$ for $i \neq j$ and $(i, j) \notin E$ (Mateos et al., 2019; Gama et al., 2020).

This general definition includes the adjacency and Laplacian matrices as instances of its class.

**Remark 1.** According to Definition 1, the existence of an edge $(i, j) \in E$ does *not imply* a nonzero entry in the GSO, $S_{ij} \neq 0$. Hence, the correspondence between a GSO and a graph is not bijective in general.

### 3.2 parametrised GSO

We begin by defining our parametrised graph shift operator.

**Definition 2.** We define the *parametrised graph shift operator (PGSO)*, denoted by $\gamma(A, \mathcal{S})$, as

$$\gamma(A, \mathcal{S}) = m_1 D_a^{e_1} + m_2 D_a^{e_2} A_a D_a^{e_3} + m_3 I_n, \tag{1}$$

where $A_a = A + aI_n$ and $D_a = \text{Diag}(A_a \mathbf{1}_n)$ is the degree matrix of $A_a$. We denote the parameter tuple corresponding the $\gamma(A, \mathcal{S})$ by $\mathcal{S} = (m_1, m_2, m_3, e_1, e_2, e_3, a)$ consisting of scalar multiplicative parameters $m_1, m_2, m_3$, scalar exponential parameters $e_1, e_2, e_3$ and an additive parameter $a$.

The main motivation of the parametrised form in Equation (1) is to span the space of commonly used GSOs and indeed we are able to generate a wide range of different graph shift operators by choosing different values for the parameter set $\mathcal{S}$. In Table 1, we give examples of parameter values in $\gamma(A, \mathcal{S})$ which result in the most commonly used GSOs and message-passing operators in GNNs. Unlike the GSO, the PGSO uniquely identifies the graph it corresponds to, i.e., the GSO and PGSO do not share the property discussed in Remark 1.

Although we base the definition of $\gamma(A, \mathcal{S})$ on the adjacency matrix, we can define the PGSO using other graph representation matrices, such as the non-backtracking operator $B$, $\gamma(B, \mathcal{S})$, (Krzakala et al., 2013; Bordenave et al., 2015) or the diffusion matrix $S$, $\gamma(S, \mathcal{S})$ (Klicpera et al., 2019).

Table 1: Known Graph Shift Operators as parameter choices $\mathcal{S}$ in $\gamma(A, \mathcal{S})$.

| $\mathcal{S}=(m_1, m_2, m_3, e_1, e_2, e_3, a)$ | **Operator** | **Description** |
|---|---|---|
| $(0, 1, 0, 0, 0, 0, 0)$ | $A$ | Adjacency matrix and Summation Aggregation Operator of GNNs |
| $(1, -1, 0, 1, 0, 0, 0)$ | $D - A$ | Unnormalised Laplacian matrix $L$ |
| $(1, 1, 0, 1, 0, 0, 0)$ | $D + A$ | Signless Laplacian matrix $Q$ (Cvetković & Simić, 2010) |
| $(0, -1, 1, 0, -1, 0, 0)$ | $I_n - D^{-1}A$ | Random-walk Normalised Laplacian $L_{rw}$ |
| $(0, -1, 1, 0, -\frac{1}{2}, -\frac{1}{2}, 0)$ | $I_n - D^{-\frac{1}{2}}AD^{-\frac{1}{2}}$ | Symmetric Normalised Laplacian $L_{sym}$ |
| $(0, 1, 0, 0, -\frac{1}{2}, -\frac{1}{2}, 1)$ | $D_1^{-\frac{1}{2}}A_1D_1^{-\frac{1}{2}}$ | Normalised Adjacency matrix of GCNs (Kipf & Welling, 2017) |
| $(0, 1, 0, 0, -1, 0, 0)$ | $D^{-1}A$ | Mean Aggregation Operator of GNNs (Xu et al., 2019) |

### 3.3 SUGGESTED METHOD: GNN-PGSO AND GNN-$m$PGSO

Next, we formally discuss how $\gamma(A, \mathcal{S})$ is incorporated in GNN models. Let a GNN model be denoted by $\mathcal{M}(\phi(A), X)$, taking as input a non-parametrised function of the adjacency matrix $\phi(A) : [0,1]^{n \times n} \to \mathbb{R}^{n \times n}$ and an attribute matrix (in case of an attributed graph) $X \in \mathbb{R}^{n \times d}$. Further, let $K$ denote the number of aggregation layers that $\mathcal{M}$ consists of. The *Parametrised Graph Shift Operator* (PGSO) formulation transforms the GNN model $\mathcal{M}(\phi(A), X)$ into the *GNN-PGSO* model $\mathcal{M}'(\gamma(A, \mathcal{S}), X)$. Moreover, we define the *GNN-mPGSO* model $\mathcal{M}''(\gamma^{[K]}(A, \mathcal{S}^{[K]}), X)$, where $\gamma^{[K]}(A, \mathcal{S}^{[K]}) = [\gamma(A, \mathcal{S}^1), ..., \gamma(A, \mathcal{S}^K)]$, i.e., we assign each GNN layer a different parameter tuple $\mathcal{S}^l$ for $l \in \{1, \ldots, K\}$.

**Message-passing steps and convolutions** In a spectral-based GNN (Wu et al., 2020), where the GSO is explicitly multiplied by the model parameters, it is straightforward to replace the GSO with $\gamma(A, \mathcal{S})$. However, with some further analysis $\gamma(A, \mathcal{S})$ can also be incorporated in spatial-based GNNs, where the node update equation is defined as a message-passing step. Here we illustrate the required analysis on the sum-based aggregation operator, where we sum the feature vectors $h_j \in \mathbb{R}^d$ in the neighborhood of a given node $v_i$, denoted $\mathcal{N}(v_i)$. The sum operator of the neighborhood representations can be reexpressed as: $\sum_{j:v_j \in \mathcal{N}(v_i)} h_j = \sum_{j=1}^n A_{ij}h_j$. Using this observation we can derive the application of $\gamma(A, \mathcal{S})$ in a message-passing step to be,

$$(\gamma(A, \mathcal{S})h)_i = m_1 (D_a)_i^{e_1} h_i + m_2 \sum_{j=1}^n (D_a)_i^{e_2} (A_a)_{ij} (D_a)_j^{e_3} h_j + m_3 h_i. \tag{2}$$

**Examples** The following examples highlight the usage of the $\gamma(A, \mathcal{S})$ operator:

1. In the standard GCN (Kipf & Welling, 2017) the propagation rule of the node representation $H^{(l)} \in \mathbb{R}^{n \times d}$ in a computation layer $l$ is,

$$H^{(l+1)} = \sigma\big(D_1^{-\frac{1}{2}}A_1D_1^{-\frac{1}{2}}H^{(l)}W^{(l)}\big),$$

   where $W^{(l)}$ is the trainable weight matrix and $\sigma$ denotes a non-linear activation function. The GCN-PGSO and GCN-mPGSO models, respectively, perform the following propagation rules,

$$H^{(l+1)} = \sigma\big(\gamma(A, \mathcal{S})H^{(l)}W^{(l)}\big) \text{ and } H^{(l+1)} = \sigma\big(\gamma(A, \mathcal{S}^l)H^{(l)}W^{(l)}\big).$$

2. The Graph Isomorphism Network (GIN) (Xu et al., 2019) consists of the following propagation rule for a node representation $h_i^{(l)} \in \mathbb{R}^d$ of node $v_i$ in the computation layer $l$,

$$h_i^{(l+1)} = \sigma\Big(h_i^{(l)}W^{(l)} + \sum_{j:v_j \in \mathcal{N}(v_i)} h_j^{(l)}W^{(l)}\Big).$$

Using the Equation (2) the propagation rule is transformed into the GIN-PGSO formulation as,

$$h_i^{(l+1)} = \sigma\Big(\big(m_1\,(D_a)_i^{e_1} + m_3\big)h_i^{(l)}W^{(l)} + m_2\sum_{j:v_j\in\mathcal{N}(v_i)}\epsilon_{ij}h_j^{(l)}W^{(l)}\Big),$$

where $\epsilon_{uv}$ are edge weights defined as $\epsilon_{ij} = (D_a)_i^{e_2}\,(D_a)_j^{e_3}$. Analogously, we can construct the GIN-mPGSO formulation by superscripting every parameter in $\mathcal{S}$ by $(l)$.

**Computational Cost**  Since in (1) the exponential parameters are applied only to diagonal matrices the PGSO and mPGSO are efficiently computable and optimisable. $\gamma(A, \mathcal{S})$ can be extended by using *vector* instead of scalar parameters. Although this extension leads to better expressivity, the computational cost is increased, as the number of parameters then depends on the size of the graph.

# 4 SPECTRAL ANALYSIS OF $\gamma(A, \mathcal{S})$

In this section, we study spectral properties of $\gamma(A, \mathcal{S})$ in theoretical analysis in Section 4.1 and through empirical observation in Section 4.2. The obtained theoretical results provide a foundation for further analysis of methodology involving the PGSO and allow an efficient observation of spectral support bounds of commonly used GSOs, that are instances of $\gamma(A, \mathcal{S})$.

## 4.1 THEORETICAL ANALYSIS

Here we investigate the spectral properties of our $\gamma(A, \mathcal{S})$. Throughout this section we assume that we work on undirected graphs. In Theorem 1 we show that $\gamma(A, \mathcal{S})$ has a real spectrum and eigenvectors independent of the parameters $\mathcal{S}$. In Theorem 2 we study the spectral support of $\gamma(A, \mathcal{S})$.

**Theorem 1.** $\gamma(A, \mathcal{S})$ has real eigenvalues and a set of real eigenvectors.

The proof of Theorem 1 follows directly from noticing that $\gamma(A, \mathcal{S})$, which is not symmetric in general, is similar to a symmetric matrix and therefore shares eigenvalues with this symmetric matrix (Horn & Johnson, 1985, pp. 45,60); the full proof can be found in Appendix A.1.

Being able to guarantee real eigenvalues and eigenvectors for all parameter values of $\gamma(A, \mathcal{S})$ enables practitioners to deploy our PGSO in spectral network analysis without having to replace elements of the algorithms which assume a real spectrum. As a result of Theorem 1 eigenvalue computations can be stabilised by working with the symmetric, similar PGSO used in the proof of Theorem 1.

Especially the theoretical analysis of the PGSO and algorithms involving the PGSO is aided by Theorem 1. There are several publications, where the complications and lack of results for complex valued graph spectra are discussed in the case of directed graphs (Guo & Mohar, 2017; Brualdi, 2010; Li et al., 2015). To remain within the real domain independent of the parameter choice (for undirected graphs) enables analysts to access a wide variety of spectral results, which only hold for real symmetric matrices. An example of such a theorem is Cauchy's interlacing theorem (Bernstein, 2009, p. 709), which can be used to relate the adjacency matrix spectra of a graph and its subgraphs. Hall et al. (2009) and Porto & Allem (2017) have been able to prove interlacing theorems for unnormalised, signless and normalised Laplacians. The fact that the PGSO has a real spectrum presents a first step to extend their work to potentially apply to the PGSO independent of the parameter values. However, this is only one example of a powerful theoretical result, which relies on a real spectrum and is accessible to our PGSO formulation due to the result shown in Theorem 1.

Now that the spectrum of $\gamma(A, \mathcal{S})$ has been shown to be real we study the spectral support of $\gamma(A, \mathcal{S})$. A common way to prove bounds on the spectral support of a matrix is via direct application of the Gershgorin Theorem (Bernstein, 2009, p. 293) as done in Theorem 2.

**Theorem 2.** Let $C_i = m_1(d_i + a)^{e_1} + m_2(d_i + a)^{e_2+e_3}a + m_3$ and $R_i = |m_2|(d_i + a)^{e_2+e_3}d_i$, where $d_i$ denotes the degree of node $v_i$. Furthermore, we denote eigenvalues of $\gamma(A, \mathcal{S})$ by $\lambda_1 \leq \lambda_2 \leq \ldots \leq \lambda_n$. Then, for all $j \in \{1, \ldots, n\}$,

$$\lambda_j \in \left[\min_{i\in\{1,\ldots,n\}}(C_i - R_i),\ \max_{i\in\{1,\ldots,n\}}(C_i + R_i)\right]. \tag{3}$$

The proof of Theorem 2 is shown in Appendix A.2.

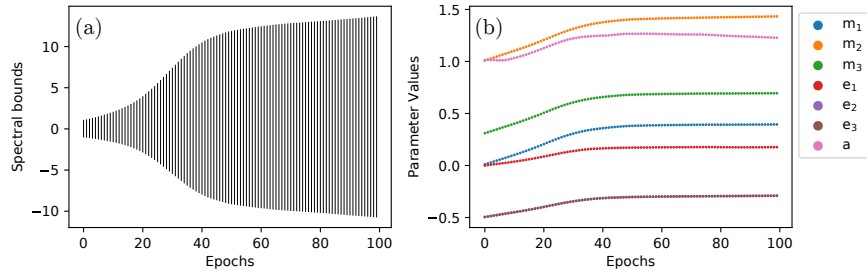

Figure 1: (a) bounds on the spectral support and (b) parameter values of $\gamma(A, \mathcal{S})$ plotted against the training epochs of a GCN-PGSO applied to a node classification task on the Cora graph. Note that the optimal values of $e_2$ and $e_3$ lie close together and hence appear as a single line in (b).

**Examples** For the parametrisation of $\gamma(A, \mathcal{S})$ corresponding to the adjacency matrix, we obtain $C_i = 0$ and $R_i = d_i$. $R_i$ is clearly maximised by the maximum degree and therefore, from (3) the spectral support of $A$ is equal to $[-d_{\max}, d_{\max}]$, as required. Similarly, the spectral supports of $L$, $L_{sym}$ and $L_{rw}$, can be deduced by plugging in the corresponding parameters into the result in (3). For the operator used in the message passing of the GCN (Kipf & Welling, 2017) a Gershgorin bound such as the one in Theorem 2 has not been calculated yet as far as we are aware. We obtain $C_i = 1/(d_i + 1)$ and $R_i = d_i/(d_i + 1)$. Therefore, from (3) the spectral support of the Kipf and Welling operator is restricted to lie within $[-(d_{\max} - 1)/(d_{\max} + 1), 1]$, the lower bound of this interval tends to -1 as $d_{\max} \to \infty$. So only in the limit is their spectral support symmetric around 0.

The bounds proven in Theorem 2 allow the observation of the spectral bounds for the many GSOs which can be obtained via specific parameter choices made in our PGSO formulation. In the context of GNNs such bounds are of value since they allow statements about numerical stability and vanishing/exploding gradients to be made. For example in Kipf & Welling (2017) the observation that the spectrum of the symmetric normalised Laplacian is contained in the interval $[0, 2]$ motivated them to make use of the "renormalisation trick" to stabilise their computations. Since the PGSO is learned its spectral support varies throughout training. The bounds in Theorem 2 enable us to monitor bounds on the spectral support in a numerically efficient manner, avoiding the computation of eigenvalues at each iteration, as is showcased in the empirical observation in Section 4.2.

Since the majority of the commonly used graph shift operators correspond to specific parametrisations of $\gamma(A, \mathcal{S})$ the spectral properties of $\gamma(A, \mathcal{S})$ are general themselves, in the sense that we cannot establish spectral results for $\gamma(A, \mathcal{S})$, which have already been shown to not be present for one of its parametrisations. This generality and lack of specific spectral features, which hold for all parametrisations is precisely one of the strengths of utilising $\gamma(A, \mathcal{S})$, since it allows the graph shift operator to manifest different spectral properties in different tasks, when they are of benefit.

## 4.2 EMPIRICAL OBSERVATION

In this section, we observe bounds on the spectral support, proven in Theorem 2 (Figure 1(a)) and optimal parameters (Figure 1(b)) of the PGSO incorporated in the GCN during training on a node-classification task on the Cora dataset. More details on the task will be provided in Section 5.3.

Surprisingly, the spectral support of the PGSO remains centered at 0 throughout training in Figure 1(a) without this being enforced by the design of the PGSO. Recall that the centre of the spectral support intervals has the following parametric form $C_i = m_1(d_i + a)^{e_1} + m_2(d_i + a)^{e_2 + e_3} a + m_3$. It is nice to observe that this desirable property of the "renormalised" operator used by Kipf & Welling (2017) is preserved throughout training. We also observe that the spectral bounds smoothly increase throughout training. The increasing support is a direct result of the learned PGSO parameters.

As we expected from our analysis of Figure 1(a), we observe the parameters of the PGSO to be smoothly varying throughout training in Figure 1(b). The parameters in Figure 1(b) can be seen to have been initialised at the values corresponding to the chosen GSO in the GCN and from there they smoothly vary towards new optimal values within the first 40 training epochs, which are then stable throughout the remainder of the training, ruling out exploding gradients. It is nice to note that

in Section 5 Figure 4(b) we observe the accuracy of the GCN using the trained PGSO parameters, displayed in Figure 1(b), to slightly outperform the standard GCN.

## 5 EXPERIMENTS

We evaluate the performance of the PGSO in a simulation study and on 8 real-world datasets. In Section 5.1, we show the ability of $\gamma(A, \mathcal{S})$ to adapt to varying sparsity scenarios through a stochastic blockmodel study, in Section 5.2 we study the sensitivity of the GNN-PGSO's performance to different $\gamma(A, \mathcal{S})$ initialisations and in Section 5.3 we evaluate the contribution of PGSO and mPGSO to the GNN performance in real-world datasets.

### 5.1 SPARSITY INTERPRETATION OF $\gamma(A, \mathcal{S})$

The choice of a graph shift operator depends on the structural information of a dataset and on the end task. For example, in graph classification tasks the datasets usually contain small and sparse graphs (Wu et al., 2018), while in node classification tasks the graphs are larger and denser. In this section, we highlight the ability of $\gamma(A, \mathcal{S})$ to adapt to different sparsity regimes.

**Stochastic Blockmodels**    In order to simulate graphs with varying sparsity levels, we utilise the parametrisation of stochastic blockmodel generation. Stochastic blockmodels (SBMs) are well-studied generative models for random graphs, that tend to contain community, i.e., block, structure (Holland et al., 1983; Karrer & Newman, 2011). Let $k$ be the number of communities, $\{C_1, ..., C_k\}$ be the $k$ disjoint communities and $p_{ij}$ be the probability of an edge occuring between a node $u \in C_i$ and a node $v \in C_j$. SBMs offer a flexible tool for random graph generation with specific properties, such as the *detectability* of the community structure (Decelle et al., 2011) and the *sparsity* level of the graph by choosing appropriate edge probabilities $p_{ij}$. Here, we focus on a restricted stochastic blockmodel parametrisation where the probabilities of an edge occurring between nodes within the same community are all equal to $p$, i.e., $p_{ii} = p \ \ \forall i \in \{1, ..., k\}$, and the probabilities of an edge between nodes in different communities are all equal to $q$, i.e., $p_{ij} = q \ \ \forall i \neq j, \ \ i, j \in \{1, ..., k\}$.

**Dataset and experimentation setup**    In this experiment, we consider 15 $p, q$ parameter combinations $(p, q) \in \{(0.5, 0.25), (0.48, 0.24), ..., (0.22, 0.11)\}$, where by decreasing the parameters $p$ and $q$ we increase the sparsity of the sampled networks. For each parameter set we sample 25 graphs with 3 communities containing 200 nodes each. Figure 5 in Appendix B.1 contains plots of adjacency matrices sampled from these 15 parameter combinations. The learning task performed by a GCN will be to assign community membership labels to each node. For all parameter choices the ratio $p/q$, i.e., the community detectability, is equal to 2, so that the theoretical difficulty the task remains constant across the different sparsity levels. As in Dwivedi et al. (2020), all of the nodes have uninformative attributes, except for one node per community that carries its community membership as its node at-

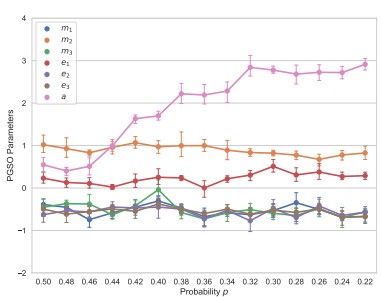

Figure 2: Mean and standard deviation of the PGSO parameters on SBM

tribute. We set the train/validation/test splits equal to 80%, 10%, 10% of nodes, respectively, for each community. We use a 3-layer Graph Convolutional Network (Kipf & Welling, 2017) with hidden size 64 and with our PGSO incorporated as the message passing operator and initialised to the GCN configuration. We, then, observe the learned parameter values of $\gamma(A, \mathcal{S})$ after 200 epochs.

**Parameter Values**    Figure 2 shows the average optimal parameter values of $\gamma(A, \mathcal{S})$ found for each one of the 15 node classification datasets. We observe that all parameters remain close to constant as the sparsity of the SBM samples increases, except for the additive parameter $a$, which can be observed to clearly increase with increasing sparsity levels. The parameter $a$ in the PGSO parametrisation plays a very similar role to the regularisation parameter of the normalised adjacency matrix, which is observed to significantly improve the performance of the spectral clustering algorithm in the task of community detection in the challenging sparse case in Dall'Amico et al. (2020)

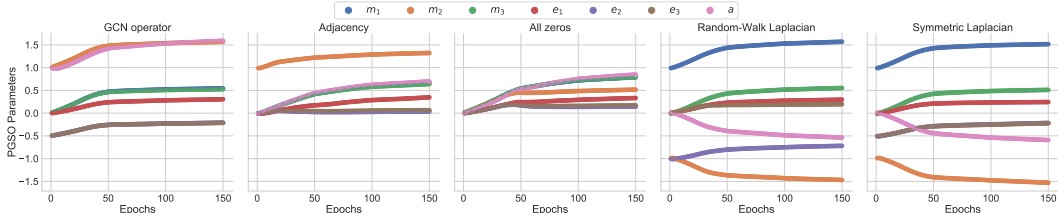

Figure 3: Parameter evolution of $\gamma(A, \mathcal{S})$ for 150 epochs, when applied on GCN-PGSO model for the node classification task on Cora.

and Qin & Rohe (2013). It is very nice to see that the PGSO automatically varies this regularisation parameter, replicating the beneficial regularisation observed in the literature, without this behaviour being incentivised in any way by the model design or the parametrisation. In Figure 2 we can furthermore observe that the $e_2$ and $e_3$ values are closely aligned for all sparsity levels indicating that a symmetric normalisation of the adjacency matrix seems to be beneficial in this task.

## 5.2 SENSITIVITY ANALYSIS OF $\gamma(A, \mathcal{S})$ TO DIFFERENT INITIALISATIONS

In Section 5.1, we initialised a 3-layer GCN-PGSO model with the original GCN parameter configuration, as shown in Table 1. In this section, we observe how sensitive the optimal parameters of $\gamma(A, \mathcal{S})$ and the final model accuracy are to different GSO initialisations.

**Experimentation setup:** We use the GCN-PGSO model, described in Section 3.3, for the node classification task on the *Cora* dataset (McCallum et al., 2000). We consider 5 different initialisations of $\gamma(A, \mathcal{S})$ that correspond to 1) the GCN operator $D_1^{-1/2} A_1 D_1^{-1/2}$, 2) the adjacency matrix $A$, 3) the random-walk normalised Laplacian $L_{rw} = I - D^{-1}A$, 4) the symmetric normalised Laplacian $L_{sym} = I - D^{-1/2} A D^{-1/2}$ and 5) a naive all-zeros initialisation, where all $\gamma(A, \mathcal{S})$ parameters are set to 0.

In Figure 3, we observe the parameter evolution of $\gamma(A, \mathcal{S})$ over 150 epochs for the different initialisations. The parameter values obtained from the GCN, adjacency matrix and all-zeros initialisations exhibit a great amount of similarity; while the normalised Laplacian initilisations lead to similar optimal values for five of the seven parameters. For all initialisations we observe that parameters $m_1, m_3, e_1, e_2, e_3$ monotonically increase until they converge. Parameters $m_2$ and $a$ initially increase for the GCN, adjacency and all-zeros initialisations, while they initially decrease for the two normalised Laplacians. The achieved accuracy of the five initialisations is plotted in Appendix B.4, where it can be observed that the resulting accuracy from the two normalised Laplacian initialisations is slightly lower than the one achieved by the remaining three initialisations. Overall, the accuracy is not very sensitive to the different initialisations.

## 5.3 REAL-WORLD SCENARIOS

In this section, we evaluate the contribution of the parametrised GSO, when we apply it to a variety of graph learning tasks. In order to highlight the flexibility of $\gamma(A, \mathcal{S})$, we perform both node classification and graph classification tasks.

**Datasets:** For node classification, we have used the well-examined datasets *Cora* and *CiteSeer* (McCallum et al., 2000; Giles et al., 1998) and *ogbn-arxiv*, that is a citation network from a recently popular collection of graph benchmarks, the Open Graph Benchmark (Hu et al., 2020). For graph classification, we have used the extensively-studied TU datasets *MUTAG*, *PTC-MR*, *IMDB-BINARY* and *IMDB-MULTI* (Kersting et al., 2016) and *OGBG-MOLHIV* dataset from OGB (Hu et al., 2020). Details and statistics of the datasets can be found in Appendix B.2.1.

**Experimentation Setup:** For the node classification datasets Cora and CiteSeer, we performed cross validation with the train/validation/test splits being the same as in Kipf & Welling (2017), while for Ogbn-arxiv, we used the same splitting method used in Hu et al. (2020), according to

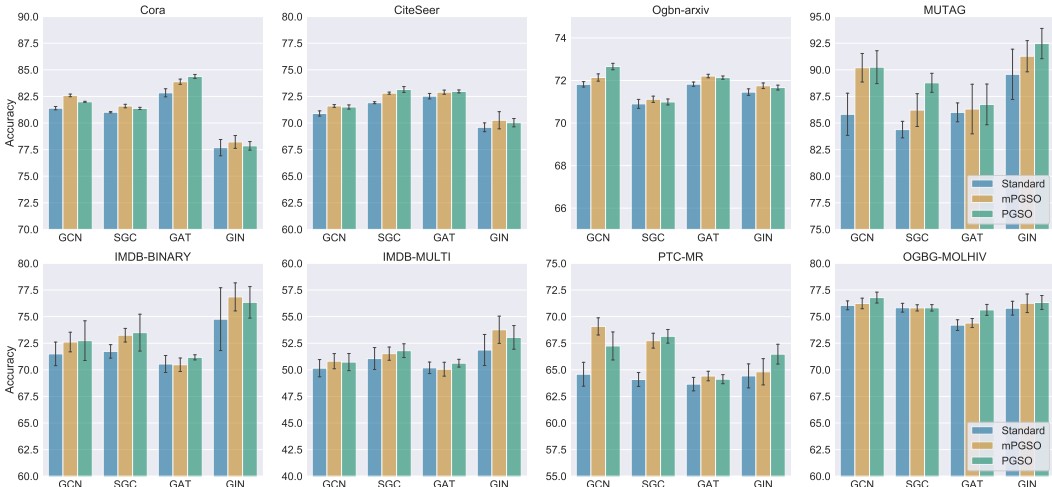

Figure 4: Classification accuracy results for both node and graph classification tasks (Validation ROC-AUC for OGBG-MOLHIV). Lower case letters denote a node classification task, while capital letters a graph classification task.

the publication dates. For the TU datasets, we performed 10-fold cross validation with grid search hyper-parameter optimisation and for OGBG-MOLHIV, following Hu et al. (2020) we used a *scaffold splitting* approach and measured the validation ROC-AUC. We compared the contribution of PGSO and mPGSO on 4 standard GNN baselines: 1) **GCN**: Graph Convolutional Network (Kipf & Welling, 2017), 2) **GAT**: Graph Attention Network (Veličković et al., 2018), 3) **SGC**: Simplified Graph Convolution (Wu et al., 2019) and 4) **GIN**: Graph Isomorphism Network (Xu et al., 2019). A full description of the experimentation details for each task can be found in Appendix B.2.2.

In Figure 4, we show the contribution of the PGSO and the mPGSO methods, when applied to standard GNNs. For all datasets and GNN architectures, the inclusion of the PGSO or the mPGSO improves the model performance. In the graph classification tasks the performance improvement is higher than the node classification tasks. Specifically, on MUTAG, PTC-MR and IMDB-BINARY, we observe a significant improvement of the classification accuracy for all GNNs. In the comparison between PGSO and mPGSO we do not find a clear winner. For this reason, we prefer the PGSO variant, as it is more efficient and never harms the model's performance. In Appendix B.5 we study the convergence of optimal accuracy and loss values of the GNN-PGSO model in a node-classification and a graph classifcation dataset. In both experiments we observe the PGSO model to converge slightly faster and to better accuracy and loss values than its conventional counterpart.

## 6 CONCLUSION

In this work, a parametrised graph shift operator (PGSO) is proposed, that encodes graph structures and can be included in any graph representation learning scenario. Focusing on graph neural networks (GNNs), we demonstrate that the PGSO can be integrated in the GNN model training. We proved that the PGSO has real eigenvalues and a set of real eigenvectors and derived its spectral bounds, which when observed in practice show that our learned PGSO leads to numerical stable computations. A study on stochastic blockmodel graphs demonstrated the ability of the PGSO to automatically adapt to networks with varying sparsity, independently confirming the positive impact of GSO regularisation which was found in the literature. Experiments on 8 real-world datasets, where both node and graph classification was performed, demonstrate that the accuracy of a representative sample of the current state-of-the-art GNNs can be improved by the incorporation of our PGSO. In answer to our two research questions posed in Section 1, our experimental results have shown that the optimal representation of graph structures is task and data dependent. We have furthermore found that PGSO parameters can be incorporated in the training of GNNs and lead to numerically stable learning and message passing operators.

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

## APPENDIX

## A PROOFS

### A.1 PROOF OF THEOREM 1

In this appendix we prove that $\gamma(A, \mathcal{S})$ has real eigenvalues and a set of real eigenvectors. For this proof the definition of similar matrices is fundamental.

**Definition 3.** Two matrices $\Phi, \Psi \in \mathbb{C}^{n \times n}$ are *similar* via a nonsingular similarity matrix $S \in \mathbb{C}^{n \times n}$ if

$$\Psi = S^{-1}\Phi S. \qquad \triangleleft$$

Next we note that $\gamma(A, \mathcal{S})$ is similar to the symmetric matrix $D_a^{-(e_2-e_3)/2}\gamma(A, \mathcal{S})D_a^{(e_2-e_3)/2}$.

Real, symmetric matrices have a set of real eigenvalues and furthermore, possess a set of real eigenvectors. It is a known property of similar matrices that they share eigenvalues and that for a given eigenvector $w$ of $\Phi$, $Sw$ is an eigenvector of $\Psi$ (Horn & Johnson, 1985, pp. 45,60). Therefore, by the similarity relationship to a real symmetric matrix via a real similarity matrix, $\gamma(A, \mathcal{S})$ has real eigenvalues and a set of real eigenvectors. □

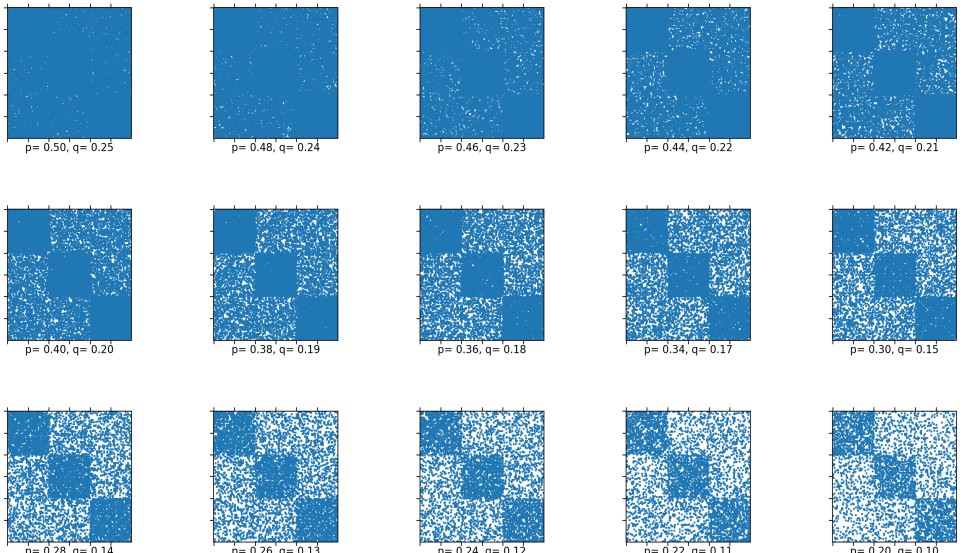

Figure 5: List of adjacency matrices generated by stochastic blockmodels for varying sparsity by decreasing the probability tuple $(p, q)$.

 PROOF OF THEOREM 2

In this proof we will again be utilising the property that similar matrices share eigenvalues by considering the matrix $D_a^{e_3} \gamma(A, \mathcal{S}) D_a^{-e_3}$, which is similar to $\gamma(A, \mathcal{S})$ . The Gershgorin Theorem states that all eigenvalues of a matrix are contained in circles centred at the matrix's diagonal elements with radii equal to the corresponding off-diagonal row-sums of the matrix elements in absolute value (Bernstein, 2009, p. 293). In Theorem 1 we showed that $\gamma(A, \mathcal{S})$ has a real spectrum and therefore the circles in the Gershgorin Theorem are in fact intervals on the real line in the case of $\gamma(A, \mathcal{S})$ . The parametrised form of $D_a^{e_3} \gamma(A, \mathcal{S}) D_a^{-e_3}$ is as follows,

$$D_a^{e_3} \gamma(A, \mathcal{S}) D_a^{-e_3} = m_1 D_a^{e_1} + m_2 D_a^{e_2+e_3} A_a + m_3 I_n. \tag{4}$$

From (4) we can simply read off that the diagonal elements, denoted $C_i$, and off-diagonal row-sums of the matrix elements in absolute value, denoted $R_i$, of $D_a^{e_3} \gamma(A, \mathcal{S}) D_a^{-e_3}$ take the following form,

$$R_i = m_1 (d_i + a)^{e_1} + m_2 (d_i + a)^{e_2+e_3} a + m_3,$$
$$C_i = |m_2| (d_i + a)^{e_2+e_3} d_i.$$

Hence, via the similarity relationship of $D_a^{e_3} \gamma(A, \mathcal{S}) D_a^{-e_3}$ and $\gamma(A, \mathcal{S})$ and a direct application of the Gershgorin Theorem applied to $D_a^{e_3} \gamma(A, \mathcal{S}) D_a^{-e_3}$ we obtain the required result. $\square$

## B EXPERIMENTS

### B.1 STOCHASTIC BLOCK MODELS

In this section, we present a visualisation of graphs generated from stochastic blockmodels (SBMs) with varying sparsity. Specifically, in Figure 5 we display 15 adjacency matrices of graphs generated from SBMs with edge probabilities $(p, q) = \{(0.50, 0.25), ..., (0.22, 0.11)\}$. The format of this adjacency matrix visualisation is taken from Dwivedi et al. (2020). The ordering of the node labels in Figure 5 corresponds to their block membership in order to highlight the block structure in the adjacency matrix plots.

## B.2 Experimentation Setup on real-world datasets

In this section, we present the dataset information and statistics and the experimentation details for the experiments in Section 5.3.

### B.2.1 Datasets

The datasets used in the experimentation setup cover both node and graph classification tasks.

- **Cora and CiteSeer** are citation networks (Kipf & Welling, 2017), where nodes correspond to documents and edges encode citation links. Both datasets contain node attributes, that are sparse bag-of-words representations of the documents.

- **Ogbn-arxiv** is a citation network with directed edges, where each node corresponds to an arXiv paper and the edges denote citations from one paper to another (Hu et al., 2020). The dataset contains node attributes, that are averaged word embeddings of the titles and the abstracts of dimensionality 128. The label of each node is the subject area of the paper and can take 40 values.

- **MUTAG and PTC-MR** are a collection of bio-informatics networks (Kersting et al., 2016), where the label of each graph correspond to a chemical property of the compound.

- **IMDB-BINARY and IMDB-MULTI** are datasets containing collaborations between actors/actresses, where each graph is an ego-graph an actor/actress and edges occur when two actors/actresses are playing in the same movie (Kersting et al., 2016). These datasets do *not* contain node attributes, thus following the literature, we create node features by using one-hot encodings of the node degrees.

- **OGBG-MOLHIV** is a collection of graphs, that represent molecules (Wu et al., 2018). Nodes are atoms and the edges correspond to chemical bonds between atoms. The graphs contain node features, that are processed as in Hu et al. (2020).

In Table 2 the dataset statistics of both node and graph classification tasks are displayed.

Table 2: Datasets statistics

| Dataset | # graphs | # avg.nodes/graph | # avg.edges/graph | # classes |
|---|---|---|---|---|
| Cora | 1 | 2,708 | 5,429 | 2 |
| CiteSeer | 1 | 3,327 | 4,732 | 2 |
| Ogbn-arxiv | 1 | 169,343 | 1,166,243 | 40 |
| MUTAG | 188 | 17.93 | 19.81 | 2 |
| PTC-MR | 344 | 14.29 | 14.32 | 2 |
| IMDB-BINARY | 1000 | 19.77 | 96.52 | 2 |
| IMDB-MULTI | 1500 | 13.00 | 76.34 | 3 |
| OGBG-MOLHIV | 41,127 | 25.5 | 27.50 | 2 |

### B.2.2 Experimentation Details

In this section, we initially report the experimental setting that is common to all experiments and, then, separately provide the details which are specific to the different tasks (node classification and graph classification).

In our experiments, we use the Adam optimizer (Kingma & Ba, 2015) with a weight decay on the parameters of $5 * 10^{-4}$ and an initial learning rate of 0.005 for the exponential parameters and an initial learning rate of 0.01 for all other model parameters. The differentiation between the two learning rates is crucial for the model training, as the fluctuation of the exponential parameters has a higher impact on the model behavior than the rest of the model parameters. We, also, used a learning rate scheduler that decayed both the learning rates by 0.5 every 50 epochs.

**Node Classification Tasks**   We report below the experimental details used in the node classification datasets, regarding model selection, metrics and model design. The Cora and CiteSeer datasets are formulated as binary class classification problems, while the Ogbn-arxiv is formulated as a 40-class classification problem.

- **Model Selection**: We perform cross-validation for all GNN, GNN-PGSO and GNN-mPGSO models with predefined dataset splits. For Cora and CiteSeer, we use the same train/validation/test splits as in Kipf & Welling (2017) for the sake of comparison with other methods that use the same splits. For Ogbn-arxiv, we use the splitting method, described in Hu et al. (2020), that splits the data into the train split, containing all papers until 2017, the validation split, containing all papers published in 2018 and test split with the published papers since 2019.

- **Hyper-parameter Tuning**: We use *grid search* to tune the hyper-parameters of the models (dimensionality of hidden units, number of GNN layers, number of epochs and the batch size).

- **Evaluation Metric**: For all node classification datasets (Cora, CiteSeer, Ogbn-arxiv), we use as evaluation metric the validation accuracy.

- **Model Design**: The number of hidden units that was grid-searched was within $\{16, 32, 64\}$. The number of GNN layers that were used were within $\{1, 2, 3, 4\}$.

- **Experiment Design**: The number of epochs for the model training was $\{50, 100, 200\}$, batch size within $\{16, 32, 64\}$ and a dropout ratio of 0.5.

**Graph Classification Tasks**   In the same fashion we report below the details of the experimentation setup of the graph classification tasks. The MUTAG, PTC-MR, IMDB-BINARY and OGBG-MOLHIV are formulated as binary class classification problems, while IMDB-MULTI is formulated as a 3-class classification problem.

- **Model Selection**: We perform 10-fold cross-validation for all GNN,GNN-PGSO, GNN-mPGSO models. For MUTAG, PTC-MR, IMDB-BINARY and IMDB-MULTI, we have used k random and stratified splits to provide balanced train/validation/test sets. For OGBG-MOLHIV we use the scaffold splitting as in Hu et al. (2020), that seperates the graphs based on their 2D structural representations.

- **Hyper-parameter Tuning**: We use *grid search* to tune the hyper-parameters of the models (dimensionality of hidden units, number of GNN layers, number of epochs and the batch size).

- **Evaluation Metric**: For the MUTAG, PTC-MR, IMDB-BINARY, IMDB-MULTI datasets, we use as evaluation metric the standard classification accuracy, while for the OGBG-MOLHIV we use as Hu et al. (2020) the validation ROC-AUC.

- **Model Design**: The number of hidden units that was grid-searched was within $\{16, 32, 64\}$. The number of GNN layers that were used were $\{1, 2, 3, 4, 5\}$.

- **Experiment Design**: The number of epochs for the model training was $\{100, 200, 300\}$, the batch size within $\{16, 32, 64\}$ and the dropout ratio of 0.5.

## B.3   TABLES

Next, we present Tables 3 and 4 containing the detailed experimentation results. In Table 3, the average accuracies with their corresponding standard deviations for the node classification tasks are presented, in particular for the datasets Cora, CiteSeer and Ogbn-arxiv. In Table 4, the results for the graph classification tasks are presented, specifically MUTAG, PTC-MR, IMDB-BINARY, IMDB-MULTI and OGBG-MOLHIV. We note that for the first 4 graph classification datasets we compute the classification accuracy, while for OGBG-MOLHIV we use the validation ROC-AUC, following Hu et al. (2020).

## B.4   MODEL PERFORMANCE FOR DIFFERENT INITIALISATIONS

In Figure 6, we show the train and validation accuracy achieved by the PGSO model discussed in Section 5.2 using the 5 different initialisation configurations: GCN operator, adjacency matrix,

Table 3: Classification accuracies for Cora and CiteSeer and validation ROC-AUC for Ogbn-arxiv.

|  | Cora | CiteSeer | Ogbn-arxiv |
|---|---|---|---|
| **GCN** | $81.4 \pm 0.4$ | $70.9 \pm 0.5$ | $71.74 \pm 0.29$ |
| **SGC** | $81.0 \pm 0.1$ | $71.9 \pm 0.2$ | $70.91 \pm 0.31$ |
| **GAT** | $83.0 \pm 0.7$ | $72.5 \pm 0.6$ | $71.82 \pm 0.22$ |
| **GIN** | $77.6 \pm 1.5$ | $69.6 \pm 0.9$ | $71.49 \pm 0.27$ |
| **GCN-PGSO** | $82.0 \pm 0.1$ | $71.5 \pm 0.4$ | $72.66 \pm 0.32$ |
| **GCN-mPGSO** | $82.6 \pm 0.2$ | $71.6 \pm 0.3$ | $72.12 \pm 0.31$ |
| **SGC-PGSO** | $81.4 \pm 0.2$ | $73.1 \pm 0.5$ | $70.99 \pm 0.28$ |
| **SGC-mPGSO** | $81.6 \pm 0.3$ | $72.8 \pm 0.2$ | $71.12 \pm 0.27$ |
| **GAT-PGSO** | $84.3 \pm 0.4$ | $73.0 \pm 0.3$ | $72.15 \pm 0.18$ |
| **GAT-mPGSO** | $83.8 \pm 0.5$ | $72.9 \pm 0.4$ | $72.21 \pm 0.17$ |
| **GIN-PGSO** | $77.8 \pm 0.8$ | $69.9 \pm 0.8$ | $71.68 \pm 0.22$ |
| **GIN-mPGSO** | $78.1 \pm 1.3$ | $70.2 \pm 1.5$ | $71.72 \pm 0.28$ |

Table 4: Classification accuracies for 5 graph classification datasets.

|  | MUTAG | PTC-MR | IMDB-BINARY | IMDB-MULTI | OGBG-MOLHIV |
|---|---|---|---|---|---|
| **GCN** | $85.8 \pm 3.1$ | $64.3 \pm 2.1$ | $71.4 \pm 2.4$ | $50.3 \pm 1.8$ | $76.06 \pm 0.97$ |
| **SGC** | $84.4 \pm 1.7$ | $64.1 \pm 1.2$ | $71.6 \pm 1.4$ | $50.9 \pm 2.0$ | $75.72 \pm 0.87$ |
| **GAT** | $86.1 \pm 2.3$ | $63.8 \pm 1.5$ | $70.8 \pm 1.5$ | $50.1 \pm 1.1$ | $74.12 \pm 0.99$ |
| **GIN** | $89.4 \pm 5.6$ | $64.4 \pm 1.9$ | $75.1 \pm 5.1$ | $52.3 \pm 2.8$ | $75.58 \pm 1.41$ |
| **GCN-PGSO** | $90.7 \pm 3.5$ | $67.1 \pm 2.2$ | $72.7 \pm 3.2$ | $51.0 \pm 1.5$ | $76.75 \pm 1.12$ |
| **GCN-mPGSO** | $90.2 \pm 2.8$ | $69.0 \pm 1.9$ | $72.4 \pm 1.8$ | $50.8 \pm 1.2$ | $76.21 \pm 1.01$ |
| **SGC-PGSO** | $88.9 \pm 1.8$ | $68.1 \pm 1.5$ | $73.2 \pm 3.5$ | $51.9 \pm 1.4$ | $75.77 \pm 0.65$ |
| **SGC-mPGSO** | $86.1 \pm 3.1$ | $67.6 \pm 1.3$ | $73.4 \pm 1.3$ | $51.5 \pm 1.2$ | $75.81 \pm 0.72$ |
| **GAT-PGSO** | $87.2 \pm 3.9$ | $64.2 \pm 0.7$ | $71.2 \pm 0.7$ | $50.6 \pm 0.7$ | $75.43 \pm 0.91$ |
| **GAT-mPGSO** | $86.8 \pm 4.3$ | $64.4 \pm 1.1$ | $70.4 \pm 1.1$ | $50.1 \pm 1.3$ | $74.23 \pm 1.11$ |
| **GIN-PGSO** | $91.9 \pm 3.1$ | $66.1 \pm 2.1$ | $76.4 \pm 2.8$ | $53.3 \pm 2.1$ | $76.32 \pm 1.37$ |
| **GIN-mPGSO** | $90.8 \pm 3.2$ | $64.7 \pm 2.2$ | $77.1 \pm 3.2$ | $53.8 \pm 2.7$ | $76.22 \pm 1.65$ |

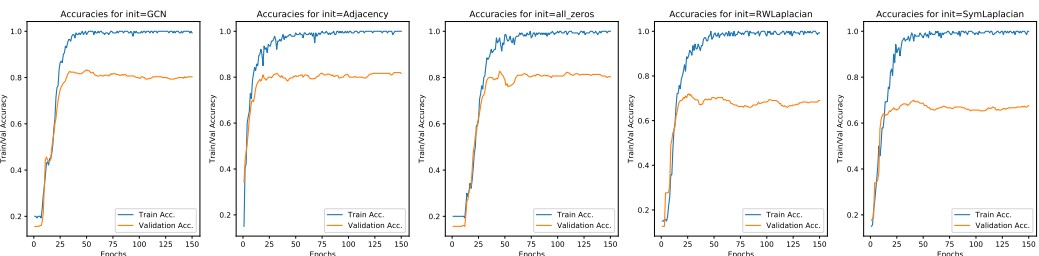

Figure 6: Train and Validation Accuracy on Cora using 5 different initialisation configurations for PGSO

all-zeros, symmetric normalised Laplacian and random-walk Normalised Laplacian. For all experiments, we used the same configuration of the hyper-parameters for a fair comparison.

## B.5 CONVERGENCE STUDY

In this appendix, we empirically analyse the contribution of the PGSO in a node classification and a graph classification task regarding the accuracy and loss convergence. For the node classification task, we have used the *Cora* dataset and a GCN model. Both for the standard GCN and the GCN-PGSO models, we used the same experimentation configuration for a fair comparison. For the graph classification task, we have used *PTC-MR* dataset and a GIN model. Both for the standard GIN and the GIN-PGSO models, we used the same configuration for a fair comparison. In Figures 7 and 8, we show the accuracy and loss convergence using the standard model and the PGSO model. As we can see, for both the node classification and graph classification task, the PGSO incorporation into the model has a positive impact on the achieved accuracy and the loss minimization throughout training. Specifically, we observe a slightly faster convergence of accuracy and loss in better values using the GCN-PGSO and GIN-PGSO models.

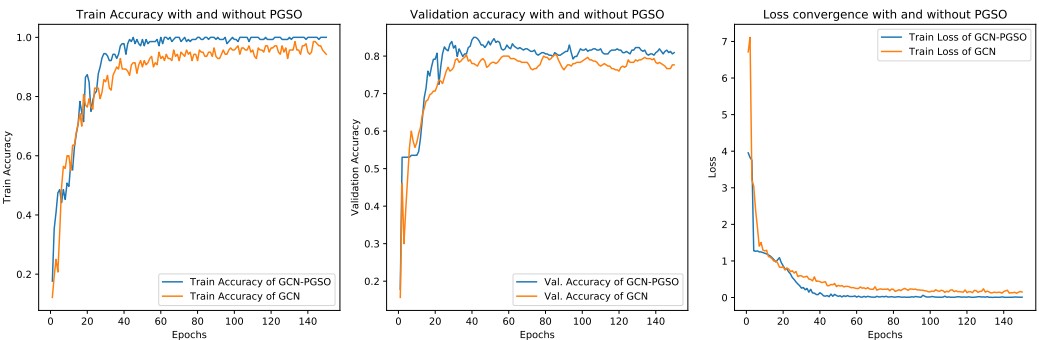

Figure 7: Accuracy and Loss convergence for the node classification task on Cora

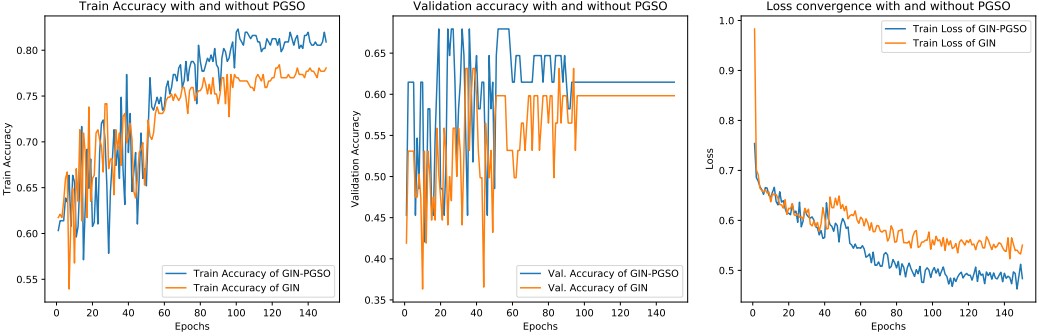

Figure 8: Accuracy and Loss convergence for the graph classification task on PTC-MR

