# OpenReview forum: "Learning Parametrised Graph Shift Operators"
_ICLR.cc/2021/Conference — ICLR 2021 Poster_

### Official Review · AnonReviewer3 · 2020-10-27
**Parametrized GSOs in GNNs**

**Rating:** 7
**Confidence:** 3

**Review:**

The paper proposes a parameterized graph shift operator (PGSO) as a replacement of fixed, hand-picked GSOs for application to GNN architectures and the experiments illustrate that PGSOs automatically adapt to the regularized GSOs for different settings. The contribution is of interest to the ML community as existing studies show significant improvements in performance of GNNs by using certain GSOs in specific contexts and therefore, a data driven characterization of GSO helps introduce adaptivity to the network representation in GNNs.

Pros: The paper is well written and easy to follow. The theoretical and empirical spectral analysis of PGSO reveals its applicability to many existing GNN architectures. The experiments are sufficient to illustrate the utility of using PGSO over a constant GSO for different settings.

Cons:
1. Remark 1 seems to contradict Definition 1. Do the authors mean 'presence' and not 'absence' of an edge in Remark 1?

2. An explicit discussion on how PGSO parameters are learned or updated in every epoch is missing.

3. The authors compare the final accuracy results of GNNS with PGSOs and those with constants GSOs. The evolution of accuracy with the number of epochs for some of these results will help evaluate the cost of learning the parameters of PGSOs.

---

> ### Author Response · Authors · 2020-11-22
> **Response to Reviewer 3: Two new experiment settings**
>
> We are very thankful for your review. Your comments have enabled us to make substantial additions to the experimental section of the paper. Specifically, we included two new experimental evaluations and a discussion of the PGSO parameter learning behaviour as well as the train/validation accuracy evolution throughout training.  You may find these additions in the revised manuscript we have uploaded in Sections 5.2 and 5.3 and Appendix B.4 and B.5. We would now like to more directly address your comments in the order that you presented them in.
>
> 1. As you correctly noticed, the term ‘absence’ in Remark 1 was a typo and has been replaced with the term ‘existence’ in the revised paper.
>
>
> 2. Our initial thought was that Sections 4.2 and 5.1 could provide information regarding the PGSO parameter behaviour on a real world dataset ( Section 4.2 and Figure 1b) and in a synthetic scenario of varying sparsity ( Section 5.1 and Figure 2 ). However, we agree with you that we should provide a more explicit analysis of the parameter evolution through epochs. For this reason, we set up another experiment on the Cora dataset using a GCN-PGSO model for 5 different initialisations (GCN, Adjacency Matrix, Symmetric Normalised Laplacian, Random-Walk Normalised Laplacian and all-zeros initial configurations) and monitored the PGSO parameters evolution over 150 epochs. For all scenarios, the $m_1,m_3,e_1,e_2,e_3$ parameters increase monotonically until they converge. Moreover, parameters $m_2$ and $a$ initially increase for the GCN, Adjacency and all-zeros initialisations, while they initially decrease for Random-Walk normalised Laplacian and Symmetric normalised Laplacian. We observe that the achieved accuracy is not very sensitive to the different initialisations. The aforementioned experimentation setup and results are included in the newly added Section 5.2 and in Appendix B.4 in the revised paper.
>
>
> 3. We thank you very much for your suggestion of the further experimentation on the accuracy evolution. Following your comments, we monitored the train/validation accuracy and the loss convergence throughout training in 2 experiments:
>     * The node classification task on Cora, where we compared the achieved accuracy and loss minimisation of the standard GCN model and the GCN-PGSO model throughout training. In both cases, we set the same hyper-parameter configurations for a fair comparison.
>     * The graph classification task on PTC-MR, where we compared again the accuracy and loss of the standard GIN model and GIN-PGSO model. Again, we used the same hyper-parameter configurations for a fair comparison.
> We observed a faster convergence of the train and validation accuracy and the cross-entropy loss to better values when using our proposed PGSO in both tasks. We included these results in Section 5.3 and the newly added Appendix B.5 in the revised paper.

---

### Official Review · AnonReviewer4 · 2020-10-27
**Not ready for publication**

**Rating:** 5
**Confidence:** 4

**Review:**

This paper describes a parameterized family of "graph shift operators", defined for any graph on n vertices as an n x n matrix where the i,j-th entry is 0 whenever edge (i,j) does not appear in the graph. The paper studies some spectral properties of the parameterized family, and experiments with using them as components of graph neural networks.

This paper appears to me to be a hammer in search of a nail. It was not clear what problem is meant to be solved here. Nor does it seem that the topic is fundamental, scientific investigation.

The paper is generally poorly written and was difficult to follow. I suspect much of that relates to the lack of clarity in the problem. However, the writing is also unfocused at the paragraph level.

In short: this paper needs substantial work before it's ready for publication.

---

> ### Author Response · Authors · 2020-11-12
> **Clarifications of the raised concerns**
>
> Dear reviewer,
> Thank you very much for your evaluation. We regret that you had to find that the paper is currently not ready for publication.  We are very much interested in improving the quality and presentation of our work. Therefore, we would like to ask you to please give us more specific feedback, which allows us to improve aspects of our work. First, please allow us to further elaborate on  the issues that you raised: the clarity of the research problem and your issues following the text structure.
>
> **Research problem:**
> The problem that we address in this work is that most of the standard graph learning models are utilising a fixed, non-learnable graph shift operator (GSO). As you correctly defined, the GSO is a matrix which encodes neighbourhood topology in graph-related tasks. We are answering the question whether there is a single GSO which optimally represents graph structures in all tasks and datasets and whether such an optimal GSO can be learned in a computationally efficient and numerically stable way. Our motivation behind this work is the observation that in different graph learning algorithms, different GSOs have been chosen without a clear motivation for this choice, e.g $D^{-1}A$ [1], $D_1^{-1/2}A_1D_1^{-1/2}$ [2], $D^{-1} L$ [3].  Our objective is to introduce a parametrised GSO that can change during the learning procedure providing a flexible data-dependent representation of the neighbourhood topology, which allows for more expressive graph learning models.
>
> **Text structure:**
> To clarify the focus of our sections and enable you to follow the content of our paper better, we provide a brief summary of the section content.
> - In Section 1 ( Introduction ) we state our two research questions. These are to firstly, find out whether there is a single graph shift operator which optimally represents graph structures in all tasks and datasets. Secondly, we aim to find out whether such an optimal representation can be learned in a computationally efficient and numerically stable way.
> - In Section 2 ( Related Work ) we firstly present examples of applications of GSOs in different ML scenarios and previous work that raised similar questions regarding the impact of choice of GSO. Secondly, we discuss examples of state-of-the-art GNNs that are using specific GSOs. Lastly, we provide a more detailed account of the work by Klicpera and highlight common themes and differences between our work and theirs.
> - In Section 3 ( Method ) we present our contribution. In Sections 3.1, 3.2, we define the parametrised GSO (PGSO) and in Section 3.3 we introduce the methods GNN-PGSO and GNN-mPGSO, where we show how we can replace the originally used GSO in GNN models with our PGSO.
> - In Section 4 ( Spectral Analysis ) we study spectral properties of our PGSO. The main objective of this Section is to show that the application of the PGSO results in numerically stable computations (as a result of the computation of the spectral bounds) and that is therefore a reliable alternative to standard and fixed GSOs (e.g a Laplacian operator).
> - In Section 5 ( Experiments ) we show empirical results of our contribution. In Section 5.1 our goal is to show how the PGSO parameters adapt to the sparsity level of a graph. In Section 5.2, we show the performance boost of standard GNN models that is achieved using the GNN-PGSO and GNN-mPGSO methods in node classification and graph classification scenarios.
>
> We hope that this summary can help address your concerns. We would be very grateful if you could provide further clarification to your review in the following areas:
>
> 1. *“It was not clear what problem is meant to be solved here. Nor does it seem that the topic is fundamental, scientific investigation.”*
> We hope that our further explanation has clarified our research problem and scientific foundation of this investigation. It would be useful if you could point out to us which aspect of the problem was lacking.
> 2. *“The paper is generally poorly written and was difficult to follow. I suspect much of that relates to the lack of clarity in the problem. However, the writing is also unfocused at the paragraph level.”*
> It would be incredibly helpful for us if you could provide specific sections of the text or aspects of the presented idea, which you thought were poorly worded or explained, so that we are able to improve the quality of this paper. We agree that the writing in the appendix can be improved and we plan to provide an improved version in our upcoming set of corrections.
>
> [1] Keyulu Xu, Weihua Hu, Jure Leskovec and Stefanie Jegelka. How powerful are graph neural networks?  In 7th International Conference on Learning Representations (ICLR), 2019.
> [2] Thomas N. Kipf and Max Welling. Semi-supervised classification with graph convolutional net-works. In 5th International Conference on Learning Representations (ICLR), 2017.
> [3] Ulrike von Luxburg. A tutorial on spectral clustering. Statistics and Computing, pp. 395 – 416, 2007.

---

> > ### Comment · AnonReviewer4 · 2020-11-24
> > **Clarifying remarks**
> >
> > >Research problem: The problem that we address in this work is that most of the standard graph learning models are utilising a fixed, non-learnable graph shift operator (GSO). As you correctly defined, the GSO is a matrix which encodes neighbourhood topology in graph-related tasks. We are answering the question whether there is a single GSO which optimally represents graph structures in all tasks and datasets and whether such an optimal GSO can be learned in a computationally efficient and numerically stable way. Our motivation behind this work is the observation that in different graph learning algorithms, different GSOs have been chosen without a clear motivation for this choice, e.g  [1], [2],  [3]. Our objective is to introduce a parametrised GSO that can change during the learning procedure providing a flexible data-dependent representation of the neighbourhood topology, which allows for more expressive graph learning models.
> >
> >
> > What you're describing here is a solution, not a problem. Prior work chooses GSOs arbitrarily: fine. What issues do we expect that to cause? What is unsatisfactory about the existing choices?
> >
> > I understand the story as this:
> > 1. there is an arbitrary choice in most graph neural net approaches
> > 2. we can instead learn the arbitrary choice
> > 3. that might do good things
> >
> > And the unclear point is: what exactly are the good things we expect? If it's mainly a speculative endeavour, so the purpose of the experiments is to find out what will happen, this should be clarified and the experiments should be designed to really isolate the effect of distinct GSOs (as opposed to, e.g., the effects of extra parameters or extensive hyperparameter search). For instance, a good start would be training multiple models that differ by fixed GSO structures and showing this induces some kind of predictable variation in their behavior.
> >
> > My read of the experiments is: learning the GSO sometimes has benefits for classification tasks. Though, it's not clear to me whether those benefits are due to a superior choice of shift operator, or simply because there are a some extra degrees of freedom in the model. The experiments showing that the learned operator is sensitivity to sparsity in the underlying seem promising, though it's not clear to me how this relates to 'optimality' of the representation.
> >
> >
> > I don't have time to give line-by-line comments on the writing---though honestly I found it less off putting on the second read. However,  a general useful principle would be to restructure the paper so that you have clear theses and the results you present are in service of each thesis. E.g., the theory section could be written as: It's useful for the GSO to have real-valued eigenvalues that lie in a known range. This is for [some clearly explained reason]. We prove the result holds.
> >
> >
> > I don't feel very strongly about this paper, so I'm not eager to be the main reason for accept or reject.  I continue to think it could benefit from extensive reworking, but I'll raise my vote to a weak accept to reflect that I'm merely lukewarm about acceptance rather than clearly opposed.

---

### Official Review · AnonReviewer1 · 2020-10-28
**An interesting idea with some promising empirical results which should be made more comprehensive**

**Rating:** 7
**Confidence:** 4

**Review:**

The authors consider the problem of learning a parametrized graph shift operator (or message passing operator) in the context of graph neural networks. They consider a family of GSO (that they name PGSO) based on seven scalar parameters, and show that it includes most commonly used operators such as the adjacency matrix or the laplacian. The spectral properties of the PGSO are analyzed. Finally, some empirical results are provided demonstrating the PGSO as a drop-in replacement for standard GSO in GNN architectures.

Graph neural networks have gathered a large amount of interest in the past years, and found widespread use in adapted problems. However, recent analyses have shown that typical architectures may have limited expressiveness (e.g. bounded by the WL test). The paper is overall well-written, and proposes a method which could be of interest to practitioners. On the other hand, the paper has some minor weaknesses, in that the proposed parametrization is redundant with some existing techniques (e.g. $m_3$ corresponds to residual connections, which are widely used), and, given the simplicity of the proposed method, a more comprehensive empirical evaluation could be beneficial. Additionally, the proposed operator is still restricted to 1-neighborhoods, and hence cannot by itself solve the expressiveness problems encountered by graph neural networks. This lack of expressiveness, in conjunction with the mixed empirical results (which show overall good performance improvement but with overlapping error bars), potentially lower the impact of this contribution.

Given the potential interest this method could hold for practitioners, I believe that the paper could be substantially improved by making the empirical evaluation more comprehensive. In particular, a more comprehensive analysis of the training dynamics (as in figure 1) as well as the effect of initialization would give a clearer picture of the effect of the PGSO in practice. Specifically, one might wonder about the importance of initialization for the $m_2$ parameter, given that it controls the influence of the edge information, and for example whether it is possible for it to change sign during training. Perhaps an informative experiment could be to initialize the PGSO at different commonly used GSO parameters (e.g. adjacency, Laplacian, normalized Laplacian), and observe how such initialization affects performance.

==============================

Edit after author response: we thank the authors for their response and providing some more empirical information. Overall I feel that this paper presents a neat idea that could be of interest to some people in the community, and I have modified my score from 6 to 7. It would be great for the authors to discuss the importance of initialization, as in particular, it seems to me that the sign of $m_2$ can never change  (from its initial value), indicating perhaps that practitioners should try initialization at either and select the better performing model.

---

> ### Author Response · Authors · 2020-11-20
> **Response to Reviewer 1: A new series of experiments**
>
>  We are very grateful for your evaluation and your constructive comments. These have allowed us to significantly improve the paper by adding a new series of experiments. These experiments provide a more comprehensive analysis of the PGSO parameter evolution throughout training and of the sensitivity to initialisation. Please find these changes in the revised manuscript we have uploaded. We now proceed by giving more detailed answers to the weaknesses you identified.
>
> > *“the paper has some minor weaknesses, in that the proposed parametrization is redundant with some existing techniques (e.g. $m_3$  corresponds to residual connections, which are widely used)”*
>
>
>
> We agree that PGSO can likely be related to elements of several different techniques; thank you very much for pointing this out. Our aim is to provide a unified view of message passing operations under the GSO formulation and as a result the strengths of several different techniques can be automatically included ad hoc by the model itself.
> Although initially it appears that $m_3$ corresponds to the residual connections, after a detailed comparison we found that there are in fact significant differences.
> The term $m_3 I_n $ of $\gamma(A,S)$ in a standard GNN model is going to be fed into a non-linear function (such as a MLP), providing, then, the node representation. Contrariwise, to our knowledge, residual connections are used to ensure that a part of the input avoids non-linear transformation, that could cause gradient explosion phenomena. So, in a setting with residual blocks, the skip connections transfer the input after the output of a GNN block, avoiding non-linear functions. In our scenario the $m_3 I_n$ term transfers the input after the aggregation of the neighborhood information, that is usually fed into a non-linear function (e.g in the Graph Isomorphism Network, the aggregated input is fed into a MLP).
> The inclusion of the trainable parameter $m_3$ in the optimisation allows for an “adaptable” incorporation of the $ m_3 I_n $ term in the model. Hence, $ m_3 I_n $ is changing through the training epochs, in contrast to residual blocks that, to our knowledge, are fixed throughout training.
>
> > *“ the proposed operator is still restricted to 1-neighborhoods, and hence cannot by itself solve the expressiveness problems encountered by graph neural networks.“*
>
> The proposed GNN-PGSO model (Section 3.3) utilises the **same** parametrised operator $\gamma(A,S)$ in all GNN layers. Hence, the PGSO parameters are **shared** through all layers and depend on information from larger neighborhoods. So while the application of the operator only propagates node features in 1-neighborhoods, the GNN-PGSO learns representations from larger neighborhoods, due to the PGSO parameter sharing between the layers. Therefore, the expressiveness of GNNs is improved by the incorporation of the PGSO. Furthermore, as we note in Section 3.2, the PGSO has the further potential to address the expressiveness problem of GNNs if, instead of the adjacency matrix, it is defined on the basis of the diffusion operator $S$ [1], that has a larger receptive field, i.e has information from larger neighborhoods.
>
> > *“the paper could be substantially improved by making the empirical evaluation more comprehensive. In particular, a more comprehensive analysis of the training dynamics (as in figure 1) as well as the effect of initialization would give a clearer picture of the effect of the PGSO in practice.”*
>
> Thank you very much for your useful suggestions on the experimental evaluation. Following your comments, we performed a series of experiments using the GCN-PGSO model in the node classification task (Cora dataset) for 5 different PGSO parameter initialisations (GCN, Adjacency Matrix,  Symmetric Normalised Laplacian, Random-Walk Normalised Laplacian initialisations and a degenerate all-zeros initialisation). For all initialisations, $m_1,m_3,e_1,e_2,e_3$ parameters monotonically increased until they converged. Moreover, $m_2$ and $a$ parameters initially increased until they converged for GCN, Adjacency matrix and all-zeros configurations, while for the normalised Laplacians, they initially decreased. Independent of the initialisations the parameters varied smoothly throughout training. Finally, it is important to note that the achieved accuracy was not very sensitive to the different initialisations and, more specifically, the accuracy from the two normalised Laplacian initialisations was slightly lower than the one achieved from the remaining three initialisations. We included these experimental results in the paper in the newly added Section 5.2 and in Appendix B4.
>
>
> [1] Johannes Klicpera, Stefan Weißenberger, and Stephan Günnemann. Diffusion improves graph learning. In Advances in Neural Information Processing Systems, 2019.

---

### Official Review · AnonReviewer2 · 2020-10-28
**A simple but useful idea**

**Rating:** 7
**Confidence:** 4

**Review:**

##########################################################################
Summary:

The paper proposes a parametric form for a matrix representation of a graph to be used as a building block within graph neural networks (GNNs). In essence, people use different normalized versions of the adjacency and Laplacian matrices within GNNs. The authors, in turn, propose a generic parametrized version that encompasses those normalizations and that can be learned from data.

##########################################################################
Reasons for score:

My overall evaluation is slightly positive. Although the methodological contribution is minor, my positive recommendation is based on the wide applicability of the simple idea proposed. The parametric graph shift operator can be readily used in existing applications of GNNs possibly boosting performance.

##########################################################################

Pros:

1. Simple contribution explained in a straightforward manner. This can help adoption by practitioners.

2. Numerical experiments rightly illustrate the value of the contribution.


##########################################################################

Cons:

1. There is no clear justification for the specific parametric form in (1), other than it can recover existing choices of the GSO. This, of course, can be obtained with other conceivable parametrizations. The authors try to justify this by saying that (1) is the "most general affine form of the adjacency matrix", but this does not seem to be a rigorous statement. Most general in what sense? What is the space of all affine forms of matrices?

2. The value of the theoretical analysis is unclear. The paragraph after Theorem 1 is supposed to highlight the value of having real eigenvalues. The authors mention "without having to worry about complex values", worrying in what sense? Why would it be difficult to implement the examples at the end of page 4 if the eigenvalues would be complex? The authors then mention the difficulty of doing spectral clustering with complex eigenvalues, but the connection with the ongoing discussion is quite loose.

3. Similarly, the value of Theorem 2 is unclear to me. Why are these bounds useful? The discussion after Theorem 2 again tries to enlighten in that direction, but it falls short.


#########################################################################

Some typos:
(1) In Remark 1, "absence of an edge" should be "existence of an edge".

#########################################################################

Edit after author response: We thank the authors for their response to my concerns and those of other reviewers. I have updated my score from 6 to 7 based on their changes.

---

> ### Author Response · Authors · 2020-11-20
> **Response to Reviewer 2: Adjustments of Sections 3 and 4**
>
> We would like to thank you very much for your careful review. Your comments have allowed us to improve the justification of the theoretical aspects of our work. We have made changes in Sections 3 and 4 to address the shortcomings which you rightfully identified and also corrected the typo you pointed out and we have uploaded a revised version of the paper. Below, we give brief responses to your main concerns:
>
> 1. As you correctly identified our motivation of the PGSO is not rigorous and tries to capture the fact that this definition includes the majority of commonly used GSOs as specific parameter choices. In the revised manuscript we have replaced the non-rigorous claim by a more specific statement of our motivation to clarify this ambiguity.
> Of course one could more formally parametrise and explore the space of affine matrix transforms of the adjacency matrix ($f(A) = m_1 A + m_0 I$) and extend this transformation by left and/or right multiplying the degree matrix raised to trainable power ($D^{e_i}$).  However, in the submitted manuscript our aim is instead to span the space of commonly used GSOs and to learn where in this space the optimal GSO lies.
>
>
> 2. We have extended the discussion of the result in Theorem 1 to address your rightfully identified shortcomings. While trying to not overstate the importance of Theorem 1, we specify that it is of both practical and theoretical significance. In practice Theorem 1 helps practitioners identify numerical errors in spectral calculations (which are not part of training a GNN). The theoretical significance of Theorem 1 stems from the fact that there exists a lot of powerful theoretical tools which apply to symmetric matrices with a real spectrum. In the revised paper, we discuss Cauchy’s interlacing theorem as an example of such a result. Through the proof of Theorem 1 it becomes apparent that these theoretical tools apply to our PGSO independent of the parameter choice and thereby Theorem 1 lays a strong foundation for further theoretical analysis of methodology, such as GNNs, making use of the PGSO. In direct response to your questions, GNNs can be implemented without issues using GSOs with complex spectra. However, theoretically there are fewer results which apply to such a GSO and hence the theoretical understanding of such models is less advanced. We agree that the link to spectral clustering here is tenuous and hence, we have removed it from this part of the discussion.
>
>
> 3. We agree that the discussion was insufficient. In Section 4.2 we demonstrate the value of Theorem 2 in an empirical study. We have now included a statement following Theorem 2, which links these sections and thereby clarifies the value of the result proven in Theorem 2. Furthermore, we have added a theoretical discussion of the value of Theorem 2. In essence, the bounds proven in Theorem 2 allow the observation of the spectral bounds for the many GSOs which can be obtained via specific parameter choices made in our PGSO formulation. In the context of GNNs such bounds are of value since they allow statements about numerical stability and vanishing/exploding gradients to be made. For example in the paper by Kipf and Welling 2017 [1] the observation that the spectrum of the symmetric normalised Laplacian is contained in the interval $[0,2]$ motivated them to make use of the “renormalisation trick” to stabilise their computations. The fact that the PGSO is learned means that its spectral support varies throughout training and the bounds proven in Theorem 2 enable us to monitor bounds on the spectral support in a numerically efficient manner, avoiding the computation of eigenvalues at each iteration.
>
>
>
> [1] Thomas N. Kipf and Max Welling. Semi-supervised classification with graph convolutional net-works. In 5th International Conference on Learning Representations (ICLR), 2017.

---

### Decision · Program_Chairs · 2021-01-07
**Final Decision**

**Decision:**

Accept (Poster)

**Comment:**

Summary:
The authors observe that a range of Laplacian-type operators used in
graph neural networks can be embedded in a parametric family, so that
the precise form of the Laplacian used can be determined by the
learning process. Empirical evaluation and some (limited) theoretical
analysis are provided.

Discussion:
The authors have provided detailed replies and also additional
experiments. That has addressed major concerns, and most reviewers now
agree the paper is good. One reviewer is more skeptical, mostly
regarding presentation. I agree with some of the points raised in this
regard, but see them as less of an issue - I would consider the
presentation improvable, but acceptable.

One weakness I should mention is that the two theorems provided are
frankly trivial. I appreciate this is 'only' a conference
submission,
but I would nonetheless call the fact that symmetric matrices have real
eigenvalues (theorem 1) an observation, not a result.
That similarly holds for any direct consequence
of Gershgorin's theorem (theorem 2). The entire page used to state this could perhaps be put to
better use for additional empirical results.


Recommendation:
The program committee (the AC and program chairs) were hesitating about this paper but decided to recommend acceptance. The idea is neat and simple, presentation and empirical evaluation are fine, if improvable (we strongly recommend the authors to invest time). What is phrased as theory is trivial, but also admittedly not the main focus of the paper.